# Iron Loss Calculation Methods for Numerical Analysis of 3D-Printed Rotating Machines: A Review

**Tamás Orosz** [1,*] , **Tamás Horváth** [1] , **Balázs Tóth** [2] , **Miklós Kuczmann** [1] **and Bence Kocsis** [3]

1. Department of Power Electronics and Electric Drives, Széchenyi István University, 9026 Győr, Hungary; kuczmann@sze.hu (M.K.)
2. Institute of Applied Mechanics, University of Miskolc, 3515 Miskolc, Hungary; mechtb@uni-miskolc.hu
3. Department of Material Science, Széchenyi István University, 9026 Győr, Hungary; kocsis.bence@ga.sze.hu
* Correspondence: orosz.tamas@sze.hu

**Abstract:** Three-dimensional printing is a promising technology that offers increased freedom to create topologically optimised electrical machine designs with a much smaller layer thickness achievable with the current, laminated steel-sheet-based technology. These composite materials have promising magnetic behaviour, which can be competitive with the current magnetic materials. Accurately calculating the iron losses is challenging due to magnetic steels' highly nonlinear hysteretic behaviour. Many numerical methodologies have been developed and applied in FEM-based simulations from the first introduced Steinmetz formulae. However, these old curve-fitting-based iron loss models are still actively used in modern finite-element solvers due to their simplicity and high computational demand for more-accurate mathematical methods, such as Preisach- or Jiles–Atherton-model-based calculations. In the case of 3D-printed electrical machines, where the printed material can have a strongly anisotropic behaviour and it is hard to define a standardised measurement, the applicability of the curve-fitting-based iron loss methodologies is limited. The following paper proposes an overview of the current problems and solutions for iron loss calculation and measurement methodologies and discusses their applicability in designing and optimising 3D-printed electrical machines.

**Keywords:** electric machines; additive manufacturing; soft magnetic materials; Preisach method; iron losses; FEM





## 1. Introduction

The additive manufacturing of Soft Magnetic Materials (SMMs) is an increasingly important area [1–5]. Within this area, the design of layered composite structures made from multiple materials has given rise to a new research field that draws parallels between laminated steel sheet cores and 3D-printed layered structures [6–9]. The following sections will refer to these materials as Soft Magnetic Layered Composites (SMLCs). The term layered structure typically refers to an alternating metal and electrically insulating material arrangement. In this paper, this terminology discusses two-component systems, but the simultaneous use of three or more materials can also occur. This technology offers exceptional design freedom, fast prototyping, and reduced material waste. For electrical machines, additive manufacturing allows designs with optimised mechanical, electromagnetic, and thermal parameters [10–12]. The proliferation of electrically powered vehicles and the limited availability of rare-earth metals demand new, advanced solutions from today's engineers [13,14]. Traditionally, iron cores for electrical machines are made from laminated steel sheets or Soft Magnetic Composites (SMCs) [15–17]. For iron cores, important physical characteristics are saturation magnetisation ($B_s$), intrinsic coercivity ($H_c$), remanent magnetic field ($B_r$), relative permeability ($\mu_r$), hysteresis loss density ($P_h$), eddy current loss ($P_e$), DC bias, and yield strength. Some of the previously mentioned characteristic quantities are depicted in Figure 1 for the case of the FeCo material. Magnetic permeability describes the strongly nonlinear connection between the magnetic field and these ferromagnetic

materials. Moreover, this hysteresis curve's shape can strongly depend on the frequency of the alternating magnetic field [18]. Figure 1 also compares the hysteresis characteristics of different 3D-printed materials, a FeCo, a FeNi, and a FeSi alloy, to an M270-50A grade, laminated steel sheet, which is a widely used material in electrical machines. The FeCo alloy has the widest hysteresis curve, which means that this material has the highest iron losses. In the case of the M270 material, there is no stress nor structural anisotropy. The FeSi material's characteristics make it the best competitor with the M270 material. A heat treatment can improve the 3D-printed material's parameters. However, this picture shows their current built-in state.

On the other hand, soft magnetic composites are suitable for forming spatial flux paths. In many cases, these can be created during the pressing process without mechanical machining so that the properties of the finished product and the starting material are similar. An additional advantage of these materials is that eddy current losses are lower, even at higher excitation frequencies than for laminated cores. However, in addition to the numerous advantages, there are disadvantages in using SMCs, with higher hysteresis losses and intrinsic coercivity than laminated iron cores and a lower relative permeability and yield strength [19]. The production of magnetic materials by metal 3D printing opens up the possibility of creating complex geometries from alloys that are difficult or impossible to machine by other methods (high-silicon steel [20–22], amorphous [23–25]).

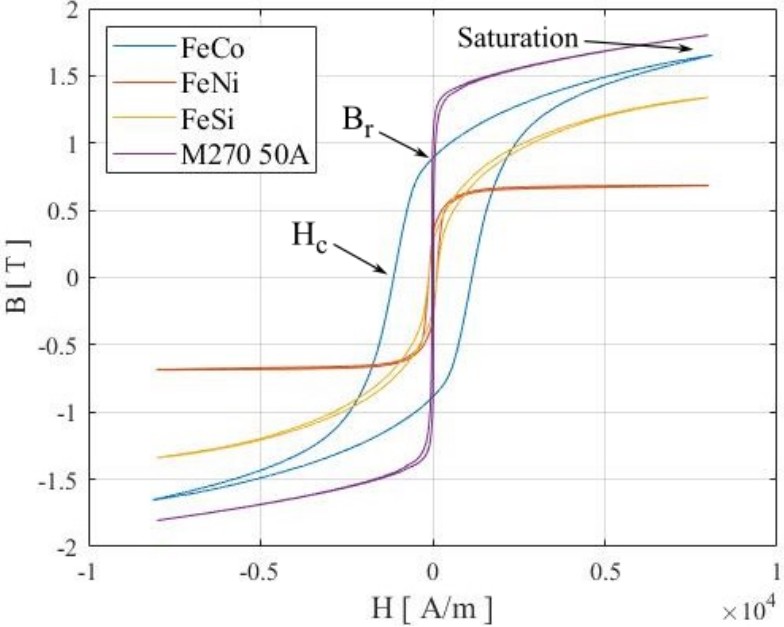

**Figure 1.** Comparing the hysteresis loops of 3D-printed toroid cores from FeCo, FeNi, and FeSi alloys with a conventional M270-50A material [26,27].

In the case of laminated iron cores, mechanical and thermal treatments can significantly change the magnetic properties of the laminated structure compared to the starting material [28]. Besides the significant amount of waste generated during lamination, another major problem with this technique is that the design of 3D flux paths is not feasible or can only be achieved by very complex manufacturing techniques. A plate thickness of a few tenths of a millimetre is common for classically produced laminated iron cores. They are fixed together by mechanical sheet forming or welding. The additive manufacturing (AM) process typically builds up the 3D part from layers of a few tens of micrometres. The layers are often fused together. In the latter case, the lower limit for the thickness of the laminated layers is always determined by the layer thickness used [29].

This paper briefly overviews the possibilities and current problems with the 3D printing of soft magnetic materials as an iron core. The paper can be divided into two main parts, where the first part of the paper shows a current overview of the leading 3D printing

technologies and the brittleness and applicability of FeSi materials in 3D-printed electrical machines. The second part overviews the iron loss calculation methodologies. Most of the proposed numerical methods can be used to model the iron losses in 3D-printed materials. These methods differ not only in their accuracy and computational demand, but they can require different complexities in measurements to determine the material-dependent parameters, as well. Moreover, the accuracy of the applied formula can depend on the type of examined material. Some of the presented formulas can be easily applied to calculate ferrite cores, but other formulas have better results for grain-oriented materials [30].

The paper shows the main applicable formulas with their advantages and disadvantages. After introducing the calculation examples, a classical and a dynamical Preisach-model-based iron loss approach are presented in a theoretical example. This profound explanation and practical example aim to help the reader select the appropriate measurement and numerical formula for the iron loss calculation task.

## 2. Advancements of 3D Printing Technology

Additive engineering of soft magnetic layered composites can result in structures with different microstructures and compositions. The selection of the appropriate printing technology (see Figure 2), the geometric design [31–33], and the variation of the process parameters and raw materials during printing [24,34,35] all offer the possibility to achieve tailored microstructures, domain structures, and magnetic properties. The fundamental goal of designing a composite structure is to reduce the magnitude of the eddy current induced in the ferromagnetic part to increase the operating frequency range, ultimately leading to the miniaturisation of our electrical equipment. If we can reduce the dissipative losses, we can improve the energy balance of our equipment by omitting cooling or reducing power. From the relation between the energy stored in the coil (where $L$ is the inductance, $I_{max}$ is the maximum current flowing in the coil) and the energy stored in the magnetic field (where $\mu_0$ is the vacuum permittivity), we can observe the following relation between the effective permeability and volume:

$$\frac{\mu_{eff}}{V} = \frac{B_{max}^2}{\mu_0 I_{max}^2 L},$$ (1)

which gives the maximum saturation of the magnetisation ($B_{max}$) for the maximal effective permeability ($\mu_{eff}$) and the minimal magnetised volume ($V$).

The different additive technologies allow the creation of different composites [36–39]. Depending on the energy source used, it is possible to form (print) the structural integrity of metals, ceramics, or polymers in the desired shape. Typically, a single apparatus can be used to deposit, melt, or cross-link a group of materials. The vast majority of studies [36,40–43] are focused on soft magnetic composites with organic insulators using simpler technologies. Here, we focused on solutions for the design of metal–ceramic composites. These research works are mainly relevant for devices capable of operating effectively in higher frequency and temperature ranges. In most cases, the printing technology used determines the composite structure that is practical or possible to produce. Based on Figure 2, Wire Direct Energy Deposition (Wire-DED), Powder Direct Energy Deposition (Powder-DED), Selective Laser Melting (SLM), and Electron Beam Melting (EBM) technologies are suitable for the formation of laminated metal–insulator composites. Of these, Powder-DED, SLM, and EBM are of practical importance. Core–shell structured SMCs can be realised by Powder-DED, SLM, EBM, Binder Jetting (BJ), and Bound Powder Extrusion (BPE) technologies [44–46].

The Wire-DED technology melts metal wires together with arc welding, quickly resulting in a preform similar to the net-shape model. The available materials are very similar to those used for welding. However, the combination of different material groups is still unresolved. The Powder-DED technology can already combine different materials or even produce metal alloys by combining elemental metal powders. At the end of printing, we obtain a near-final product. In many cases, only the machining of the fitting and contact

surfaces is necessary. SLM allows the use of pre-alloyed powders. The technology is not among the fastest processes, but its support-free printing is also available on the market. In some cases, multi-material printing is also possible. The internal stress state following the cooling is a major problem in each case. Multi-material printing significantly increases the occurrence of delamination and microcracks, which can lead to the complete failure of the printing. The stress state can be reduced during the printing process by systems supplemented with thermal imaging cameras, which can correct parameter settings layer-by-layer, or it is also possible to use a stress-relieving heat treatment post-process.

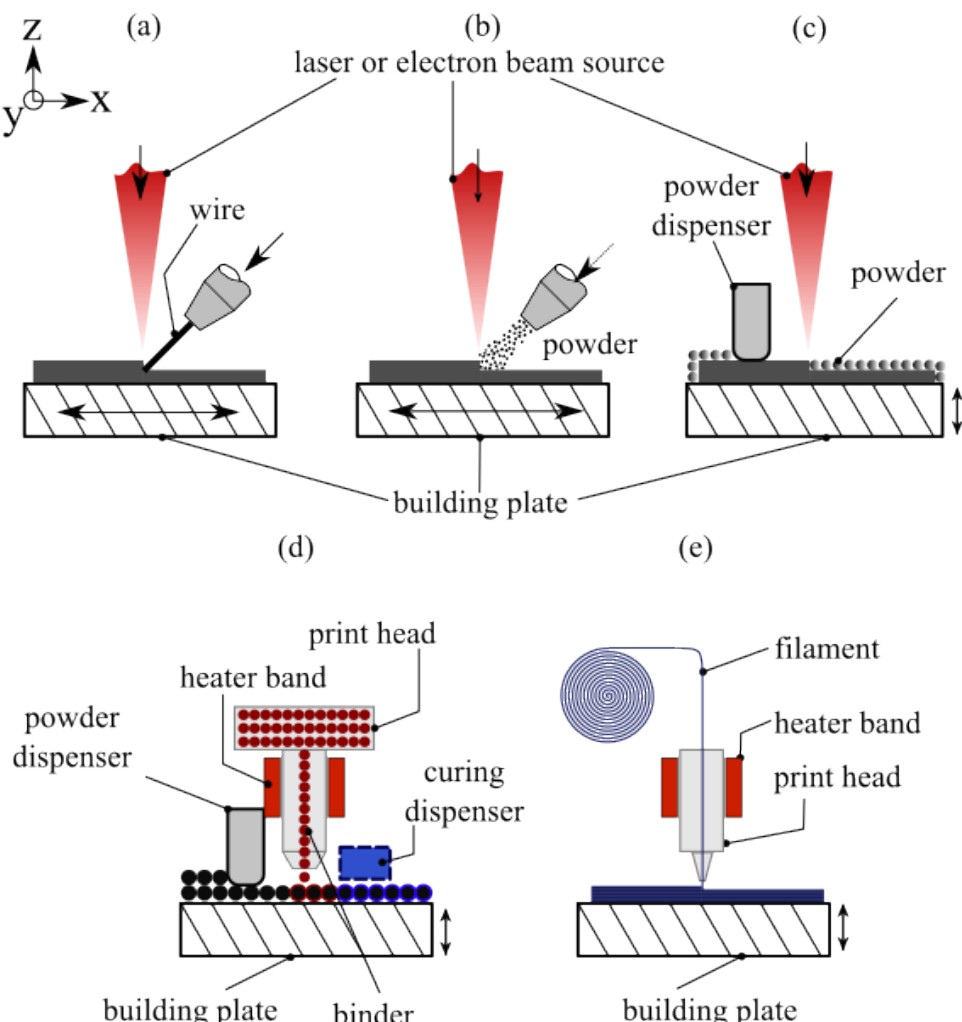

**Figure 2.** The image shows different 3D printing techniques. Wire direct energy deposition (**a**); powder direct energy deposition (**b**); selective laser melting or electron beam melting (**c**); binder jetting (**d**); bound powder extrusion (**e**).

The typical soft magnetic properties of additively manufactured structures and those of powder cores and laminated structures produced by classical manufacturing techniques differ in many ways. In addition to the internal coercivity, the static permeability and cut-off frequency values can qualify the softness of core–shell and laminated structures during measurements. The cut-off frequency can be calculated from Snoek's law [47]. The cut-off frequency is where the real part of the complex permeability changes significantly. Figure 3 summarises the characteristic values for the two types.

The coercive field of soft magnets increases with decreasing grain size. A size-dependent demagnetisation factor explains the grain size dependence of coercivity. Mager

gave the phenomenological description [48] in the form of a linear relation between the coercive field and the reciprocal of the grain size (1/GS), in which the slope is proportional to the ratio of the domain wall surface energy ($\gamma$) to saturation polarisation ($J_s$). The effect of the grain size is evident for the AM SMLC and powder cores. For layered structures, the detrimental effect of surface pining should be considered. Considering the effectiveness of these effects, we prepared the soft magnetic parameters ($H_c$, $\mu_{static}$, and $f_{lim}$) evaluation table presented in Figure 3. The dominance of eddy currents determines the limitation of the cut-off frequency. By reducing the layer thickness and preventing the formation of a large current path, it is possible to extend the operational frequency range. In the design of the composite structure, particular emphasis should be placed on the choice of the filling factor. The insulating of core–shell materials as a grain significantly increases the proportion of the insulating phase in the structure, which deteriorates the macroscopic magnetic properties of the material (e.g., $B_s$). Forming a coherent, damage-free insulating layer with a few micrometres of thickness is not feasible in 3D printing to obtain a quasi-porosity-free, mechanically sound structure after printing. In contrast, a layered design allows the deposition of a certain thickness of the insulating layer per predefined number of layers. The appropriate parameters for each type of material guarantee the printing of a quasi-compact structure. Due to the cyclic thermal stresses during production and the typically low thermal shock resistance of ceramics, it is advisable to minimise the metal–ceramic contact surfaces and to combine material pairs with nearly identical thermal expansion coefficients [35]. In addition, it is essential to note that the mechanical properties (except compressive strength) of ceramic layers are far below those of pure metallic structures [6,35,49,50].

| Microstructure of SMCs | | $H_c$ | $\mu_{static}$ | $f_{lim}$ |
|---|---|---|---|---|
| AM core-shell | | high | small | high |
| AM layered | | small | medium | high |
| PM powder | | high | medium | high |
| Laminated | | small | high | small |

**Figure 3.** The most-important soft magnetic properties of SMCs with different microstructures.

It is also important to mention the possibility of using additively manufactured SMC iron cores with a sinterless powder or air layer for electrical insulation [6,21]. In these cases, the ferromagnetic filling factor could be better since a continuous air layer can only be

formed by not sintering several layers, resulting in an insulating layer with a thickness of several hundred micrometres [21]. The dust particles trapped in the air gap create electrical percolation paths between the layers, which increases eddy current losses. An advantage, however, is that there is only a single base material, so there is no mixing of two powders with different compositions during the printing process of the layered structure. From an economic point of view, it is significant that the subsequent separation of different raw materials is often not feasible or only very costly [51]. It is important to note that one of the most-expensive components of powder additive technologies is currently the gas-phase powder raw material.

Regarding technological implementation, the Wire-DED process has yet to show practical relevance for producing metal–ceramic composites. In the Powder-DED and EBM processes, it is relatively easy to alternate metallic and ceramic materials by employing a suitable adjustment of the powder dosage and the applied energy density [52–63]. In the SLM process, the applied laser wavelength differs by three orders of magnitude for metallic and ceramic materials. A dual-laser system can be used for printing, including a laser with a wavelength of 1 μm for melting metallic material and a laser with a wavelength of 1 mm for ceramics. However, this problem can be eliminated by, for example, in situ oxidation during printing, nitriding, carbonisation, or possible post-printing annealing processes [24]. There are ceramics with absorption coefficients close to those of certain metallic materials, so direct metal–ceramic composite printing cannot be ruled out [34,50]. Printing the metal–metal structure and its subsequent post-process modification (nitriding, carbonisation, oxidation) can promote better interfacial contact between the two base metals and result in defect-free microstructures and macrostructures.

In many ways, additively manufactured iron core geometries offer engineers new design perspectives. It is also important to note that most metal 3D printing technologies impart a unique microstructure to the printed structure. In the case of SLM technology, a part built layer-by-layer in the Z direction on a table plate placed in the XY plane develops columnar grains parallel to the Z axis [22,64]. For sheet metal bodies, it is feasible to have the magnetic flux lines and the columnar grain structure of the metal plate coincide, which is half of the easy magnetisation direction for silicon steels. On a theoretical plane, in the case of toroidal iron cores, it would be possible to distort the crystal structure along the circumferential arc to a certain extent. Oliveira et al. investigated the build orientation effect on the magnetic properties of managing steel 300. They concluded that there is likely a correlation between the residual stresses and coercivity, remanence. The ring-like specimens' printing orientation were XYZ, XYZ-45°, and YZX. Figure 4 shows the magnetisation curves and hysteresis loops of these samples. M. Garibaldi et al. examined the grain orientation and structure for different print orientations, which are illustrated in Figure 5, for Fe-6.5wt %Si composition.

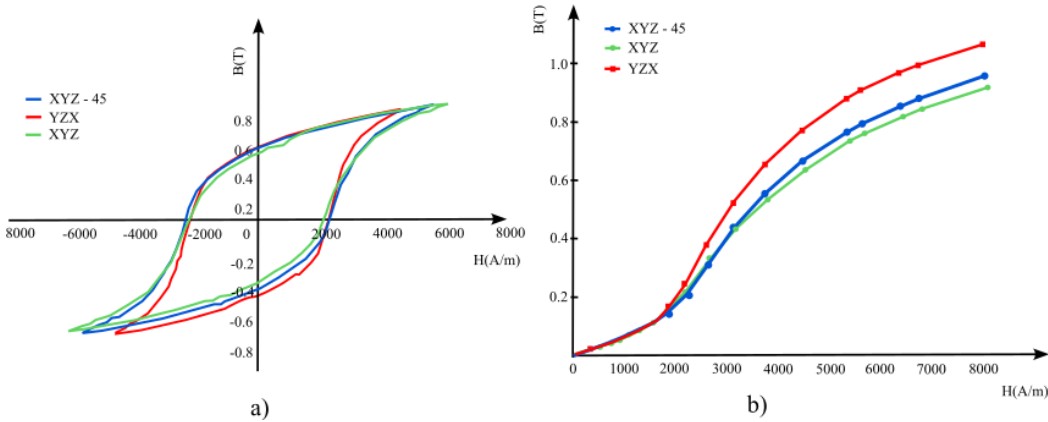

**Figure 4.** Magnetization curve of additively manufactured toroidal managing steel 300 specimens (**b**) and its BH loops at 0.5 mHz (**a**) [65].

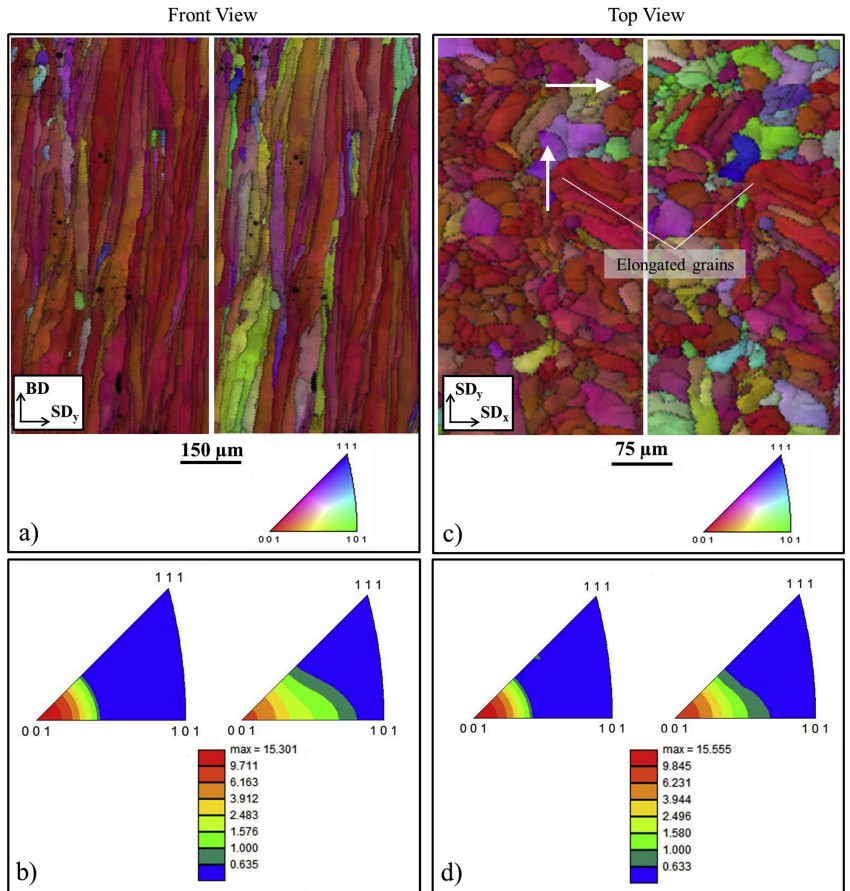

**Figure 5.** The 3D-printed high-silicon steel parts' EBSD maps and IPFs. The left (**a**,**b**) subfigures shows the texture along the build direction, while the (**c**,**d**) shows the texture in the XY plane, which is parallel to the build plate [22].

The raw materials most-commonly used in producing additively manufactured iron cores are the same as those used in the classical production of components. There are several kinds of research works on macroscopic sintering of amorphous and partially crystalline alloys, which have been popular since the 1980s [23,66–69]. The difficulty of the realisation is that rapid cooling is not possible, as the melting and sintering of new layers is cyclically performed during the printing process. The possibility of rapid cooling may still work for the first layers, but the internal stresses that freeze can cause significant mechanical and magnetic degradation during subsequent use. With expensive nucleation inhibitors and nucleation-inhibiting or stimulating alloying agents, a specific size range can be achieved, resulting in an amorphous/partially crystalline structure. The problems described above can be avoided using technologies such as Fused Filament Fabrication (FFF = BPE) [70]. In our research, we guaranteed the amorphousness and low cost of the raw material for SLM technology by using simple iron metalloids (FeC, FeB, FeSi, FeP) with significantly different atomic diameters, preventing the Short-Range Ordering principle (SRO principle) [23]. The atoms do not have time to arrange themselves in a crystalline structure due to the size difference and the thin layer application. The large-sized steel base plate on which the printing takes place has enough heat dissipation capacity to achieve quasi-rapid cooling. The 3D printability of high-saturation-magnetisation $Fe_{35}Co_{65}$ or the extremely soft $Fe_{20}Ni_{80}$ alloy, known under the Permalloy brand name, is a relatively easy task since standard steel powders often contain significant amounts of Co and Ni in addition to Fe [64,71–73]. The printability of Fe-Si alloys becomes problematic somewhere around 6.5wt% silicon [74]. The presence of high amounts of silicon causes the material to become extremely brittle, which can easily crack due to cyclic thermal stress and internal stresses during printing. By reducing the difference between the sintering and table

temperatures, i.e., by significantly increasing the table temperature, microstructural defects can be somewhat eliminated, and large complex geometries can be formed from this material. In addition, process parameters such as laser power, scanning speed, focal point diameter, and scanning strategy all impact the resulting macrostructure and its magnetic properties [75,76].

There are several kinds of research on the printability of Fe-6.5wt%Si with zero magnetostriction, which is a comparatively inexpensive material, but has relatively good soft magnetic properties [9,22,38,76]. The porosity values for pre-alloyed specimens made of gas-atomised powder were investigated for different values of volumetric energy density. M. Garibaldi et al. investigated the effect of the laser printing parameters on the microstructure and crack formation. It has been shown that irregularly shaped porosities typically form at the interfaces of shallow and wide melt pools, while spherical pores form at the bottom of deeper melt pools. They also proved that the cumulative crack length (c.c.l.) parameter starts to increase significantly above an energy input of 280 J/m. However, large, irregularly shaped pores can also be eliminated from the microstructure in the case of an energy input of 280 J/m. These results are shown in Figure 6.

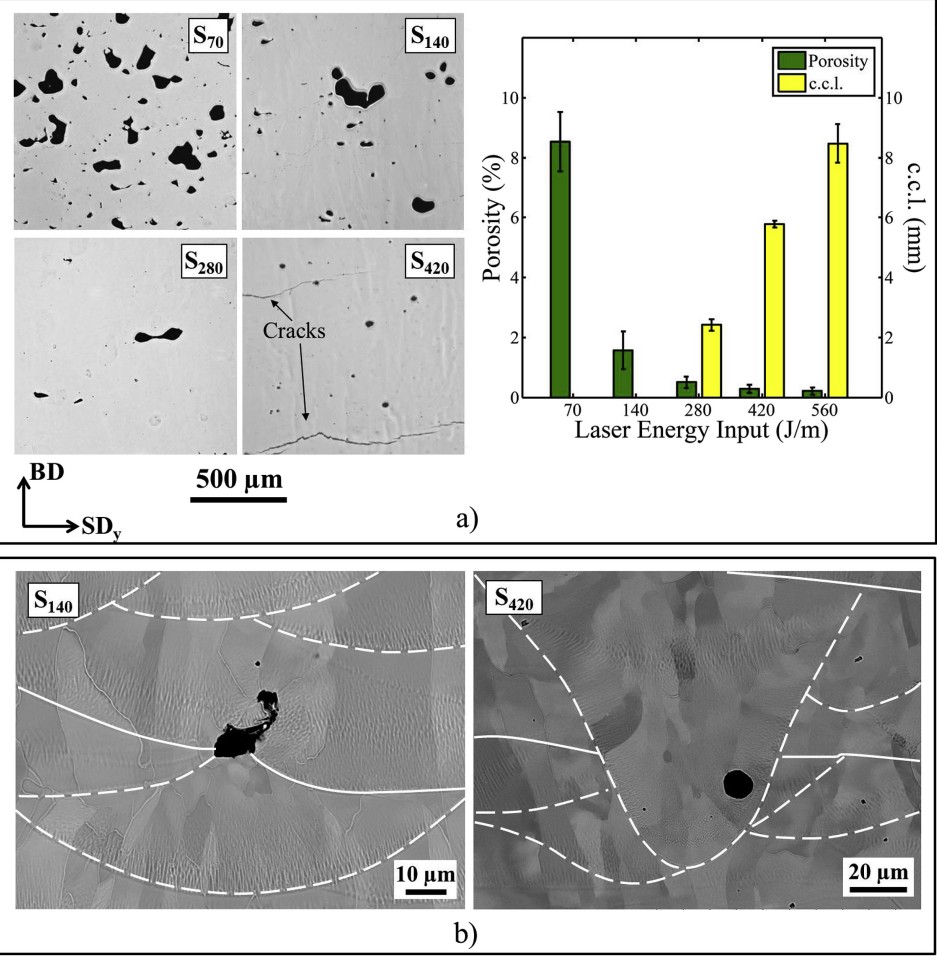

**Figure 6.** Relationship between laser energy input and sample porosity. In (**a**), optical images of SLM samples are shown (left) and the trends of the laser energy input, porosity, and cumulative crack length trends (right). (**b**) shows SEM micrographs of irregular (left) and spherical pores (right) [22].

A solution to avoid structural defects and high brittleness is the development of a gradient composition, which can be obtained by appropriate layering and laser sintering of elemental iron and silicon powders. In this case, the ideal alloying amount of 6.5wt%Si will be present only in a certain cross-section, with more or less silicon being observed elsewhere. The cross-sectional gradient changes of the elemental composition are shown

in [26]. The CT porosity measurements and scanning electron microscopy sectional view of such a sample is illustrated in Figure 7.

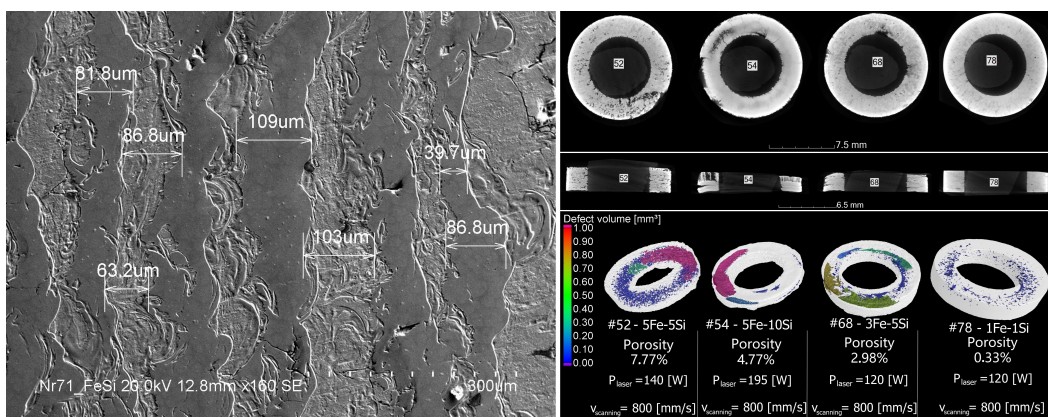

**Figure 7.** SEM micrograph of 3D-printed Fe-Si layered structure (**left**) and its reconstructed CT images (**right**) [35].

### 3. Measurement Methodologies for Iron Losses

When measuring the magnetic properties of 3D-printed parts, it is important to remember that the manufacturing technology, the geometry of the printed part, its location in the printing space, and its material composition all influence the selection of the appropriate measurement method. In the measurement technique of 3D-printed structures, we must distinguish between the measurement of sheet metal and complex geometries (toroid, stator, rotor, etc.). The relationship between printing and measurement parameters must be taken into account in all cases. In classic manufacturing processes, such as rolling Fe-Si sheets, both oriented and non-oriented grain structures can be formed. A similar situation exists for 3D-printed materials, with the difference that, in the case of PBF technologies, a dendritic grain structure forms in the direction of layer-by-layer construction (referred to as the direction along the Z axis). It is, therefore, easy to understand that, in the case of the 3D printing of sheet specimens, a permanent grain orientation can be established, which, under appropriate measurement setup, aligns with the direction of magnetic flux. Further parallels can be drawn between the cutting techniques for sheets (such as stamping, punching, abrasive or laser cutting, wire electric discharge machining) and laser sintering, as the microstructure and domain structure that form in both cases can often be modified, made unique through the appropriate selection of the cutting parameters and laser scanning strategy.

There are several standardised methods for the measurement of laminated iron cores. These standards precisely define the measurement conditions, the shape and physical dimensions of the sample, as well as the parameters of the instruments required for the measurement. The IEC standards [77–79] distinguish three different measurement methods based on the shape of the measured sample. These are the so-called Epstein frame, single-sheet, and toroidal core measurement methods [80]. In the case of 3D-printed iron cores, there are no standardised measurement procedures; researchers usually use the toroidal core measurement method [81,82].

All of the measurement methods presented below are computer-controlled, and their general principle of operation is shown in Figure 8. For the measurement, a primary ($N_1$) and a secondary ($N_2$) coil need to be placed on the sample. The primary coil should be connected in series with a resistance ($R$) without inductance and fed with a sinusoidal voltage, while the secondary circuit should be left unloaded. This essentially creates a transformer operating in an open-circuit condition. The voltage across the resistance ($u_1$) and the voltage across the open secondary coil ($u_2$) are measured using a data acquisition card. The computer processes the voltage measurements from the data acquisition card and performs the necessary calculations. According to the standards, the tested samples need

to be demagnetised before the measurement by applying an excitation that corresponds to more than ten-times the coercive field strength and gradually reducing it to zero.

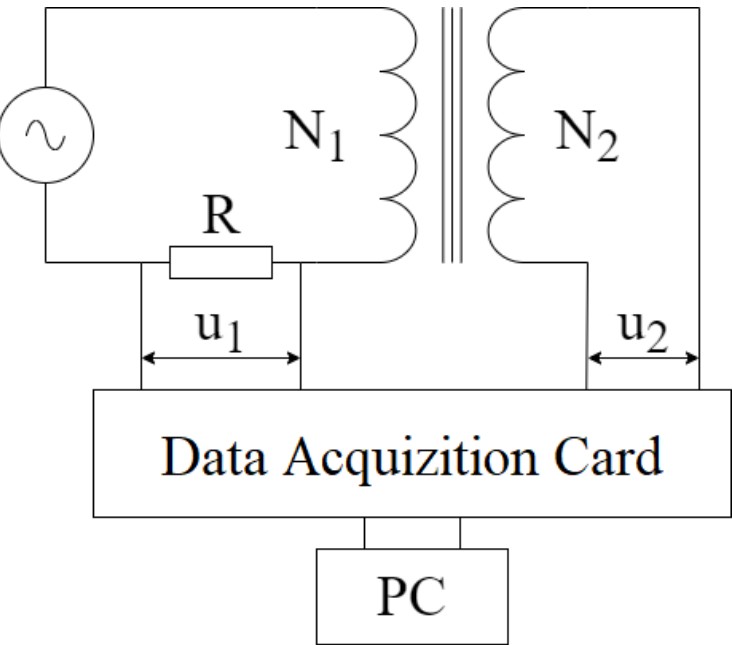

**Figure 8.** Block diagram of the computer-controlled iron core loss measurement process.

Using the measured voltages $u_1$ and $u_2$, the magnetic field strength ($H$), magnetic induction ($B$), and power loss ($p_s$) can be calculated using the following equations [83]:

$$H = \frac{N_1}{Rl_{eff}} u_1(t), \tag{2}$$

$$B = \frac{1}{N_2 A} \int_0^t u_2(\tau) d\tau, \tag{3}$$

$$p_s = \frac{fN_1}{N_2 mR} \int_0^T u_1(t) u_2(t) dt, \tag{4}$$

where $N_1$ is the number of turns in the primary coil, $N_2$ is the number of turns in the secondary coil, $m$ is the mass of the sample, $l_{eff}$ is the effective length of the magnetic field, $A$ is the cross-sectional area of the sample, $R$ is the purely ohmic resistance, $f$ is the excitation frequency, and $T = \frac{1}{f}$ is the period of the waveform.

*3.1. Epstein Frame*

In the case of the Epstein frame [84] measurement method, the laminated iron cores to be examined are arranged in a square configuration, as shown in Figure 9a. The edges of the square consist of multiple laminated iron core strips, and double-overlapping is applied at the corners so that the cross-sectional area and average length of the frame are equal on all sides of the frame. The strips used for the frame sides are cut from a rolled sheet coil to produce the iron core. The cutting direction can be the same as the rolling direction at each side of the frame. However, different cutting directions can also be chosen to measure anisotropy. Th 3D-printed iron core strips can be cut from the printed material to construct the frame [85]. Primary and secondary winding arrangements are placed on each side of the frame. The four primary and secondary windings are connected in series accordingly, forming a common primary and secondary winding. The standard defines the dimensions of the frame, the minimum weight, and the number of turns for the windings.

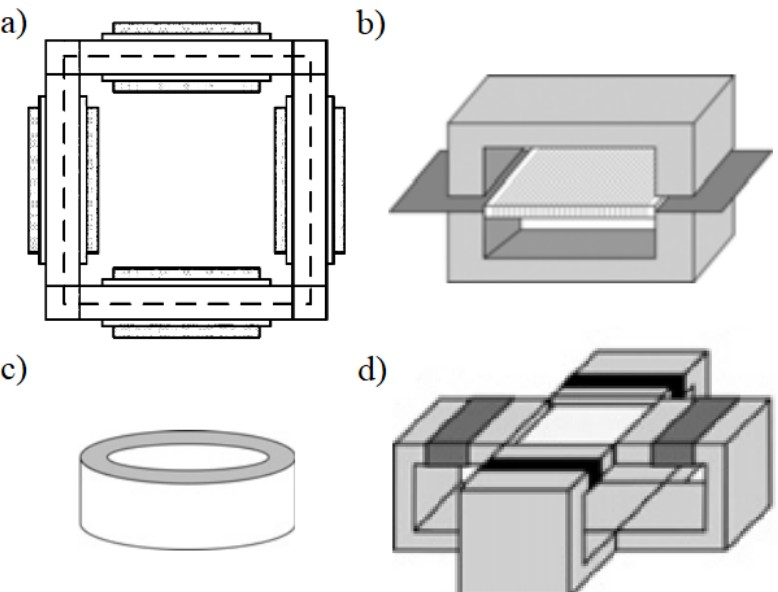

**Figure 9.** Methods for measuring iron core losses. Epstein frame (**a**); single-sheet tester (**b**); toroidal sample (**c**); multidimensional method (**d**) [83].

### 3.2. Single-Sheet Tester Measurement

In the case of the single-sheet tester [86] measurement method, the sample under investigation consists of a single sheet. The primary and secondary windings are placed on the sample, and two steel yokes enclose the magnetic circuit. These yokes are made of high-quality electrical steel, typically grain-oriented electrical steel or nickel-iron alloy, to minimise their influence on the measurement results. The standard defines the dimensions of the sample and the steel yokes, the magnetic properties of the steel yokes, and the number of turns for the windings. The fundamental structure and realisation of the single-sheet tester method can be seen in Figure 9b.

### 3.3. Toroidal Sample Measurement

The properties of magnetic materials can also be measured using toroidal samples, as shown in Figure 9c. According to international standards [79], this method is suitable for examining special alloys, nanocrystalline and amorphous materials, injection-moulded and cast materials, pressed and sintered materials, as well as soft magnetic composite materials, but not magnetic laminations. This also includes 3D-printed toroidal samples. For the measurement, toroidal samples can be created in various ways, such as using wound strips, rings cut from larger sheets, pressed powder, cutting cores from larger materials, or 3D printing toroidal cores. It is essential to mention that the measurement results depend on the sample preparation [80]. For example, windingintroduces stress in the material, which affects the magnetic parameters of the material. The primary and secondary windings are placed on the toroidal sample. For toroidal samples, the standard does not specify the exact geometrical dimensions or the number of turns in the windings; it only provides some recommendations. Moreover, this method is not suitable for measuring the magnetic properties of the material in a specific direction, thus for investigating grain-oriented cores.

### 3.4. Multidimensional Measurement Methods

The examined sheet is excited in one direction in the previously discussed methods. However, in the case of rotating magnetic machines, the iron core is subjected to magnetisation in multiple directions simultaneously. In addition to alternating current excitation, rotating magnetisation also occurs. The losses in the iron core vary depending on the magnitude of rotating and alternating excitations. In the case of moderate amplitudes, the losses arising from rotating magnetisation can be significantly greater. Therefore, to examine the

iron core material, it may be important to conduct multidimensional analysis [87], exposing the sample to multi-directional excitations. Multidimensional analysis is also important for materials with high anisotropy, such as grain-oriented materials.

In the case of multidimensional analysis, the magnetic field is controlled using the same steel yokes as in the single-sheet measurement method. Rotating and multi-directional magnetic excitation can be generated by rotating the steel yokes or with stationary steel yokes and appropriately controlling the excitation of the windings. The arrangement in Figure 9d allows for the acquisition of the hysteresis characteristics of the sample in the x and y directions. From that, the hysteresis loss of the sample can be calculated using the following equation:

$$p = \frac{f}{\rho} \int_0^T \left( B_x(t) \frac{dH_x(t)}{dt} + B_y(t) \frac{dH_y(t)}{dt} \right) dt. \tag{5}$$

Based on the steel yoke method presented here, three-dimensional measurements can also be performed by extending Equation (5) to include the third component along the y axis.

## 4. Iron Loss Modelling Approaches, an Overview

Calculating the magnetic losses of electrical machines is a highly engineered task, which needs to consider many factors. The iron losses depend not only on the applied material, but the manufacturing method, as well [88–90].

FEM-based solvers usually use a post-processing step to calculate iron losses. In this solving step, using the calculated magnetic flux distribution values, loss values are calculated for the element using an analytical material model [91–93]. The accuracy of these formulae usually depends on the measurements to which the free parameters of the formula are fit. These measurements can be made on an Epstein frame, a toroidal core setup, or in a single-plate tester [91], depending on which method is best suited to the particular calculation. The following section aims to introduce these iron-loss-calculation methods and describe their applicability, besides their advantages and disadvantages.

### 4.1. Steinmetz-Equation-Based Formulas

The first and simplest such material model, still widely used today, is the Steinmetz formula [94], which is fit to a measurement made by a sinusoidal excitation at a given frequency. Currently, this form is extended with a frequency-dependent parameter; this formula is referred to as the Steinmetz formula or power law as well in the literature [95–105], and it can be written in the following form:

$$p = C_{sc} \cdot f^{\alpha} \cdot \hat{B}^{\beta}, \tag{6}$$

where $p$ represents the power loss in $W/m^3$, $\hat{B}$ represents the peak value of the magnetic flux density in the examined material, $C_{sc}$ is the Steinmetz coefficient, and the $\alpha$ and $\beta$ coefficients are usually determined from the measurements. This formula works in a limited frequency range for sinusoidal waves and a constant or DC-biased excitation frequency [100,106].

The very first extension of this formula was the Modified Steinmetz Equation (MSE). The model assumes that this remagnetisation rate is proportional to the change of the magnetic flux density; a macroscopic re-magnetisation-based equivalent frequency ($f_{eq}$) can be introduced by the following formulae [107–109]:

$$f_{eq} = \frac{2}{(B_{max} - B_{min})^2 \pi^2} \int_0^T (\frac{dB(t)}{dt})^2 dt. \tag{7}$$

Inserting this $f_{eq}$ into Equation (6), we obtain the following form for the modified Steinmetz equation [107–110]:

$$p = C_{sc} \cdot f_{eq}^{\alpha-1} \cdot \hat{B}^{\beta} \cdot f_r, \tag{8}$$

where $f_r$ is the remagnetisation frequency, and the other parameters $C_{sc}$ $\alpha$ and $\beta$ are the same as introduced in (6). With an application of a second correction factor, this formula can also handle the DC bias premagnetisation; however, this formula is not accurate when the exciting wave fundamental frequency is relatively small, and this formula is not internally consistent [106,109,110].

Another approach, the generalised Steinmetz equation, considers the rate of magnetic induction change and depends on its instantaneous value. This formula is consistent, but does not fit well with the measurements if the third or higher harmonic part of the formula becomes significant; thus, it can result in multiple peaks in the resulting waveforms [99,106,110]. The improved generalised Steinmetz equation resolves these problems with the multiple waveforms [95,111], dividing the flux density waveforms into major and minor loops and calculating the iron losses separately in the following form:

$$p = \frac{1}{T} \int_0^T C_{SE} |\frac{dB}{dt}|^{\alpha} |\Delta B|^{\beta-\alpha} dt, \tag{9}$$

where $\Delta B$ defines the peak-to-peak flux density of the minor and major loops of the current waveform. Applying $\Delta B$ instead of B(t) resolves the DC bias sensitivity problem [95,110]. Another method, the natural Steinmetz equation, uses a similar change of the time-dependent flux density ($B(t)$) value with $\Delta B$ [104] to improve the performance of the approximation in a different way:

$$p = (\frac{\Delta B}{2})^{\beta-\alpha} \frac{C_{SE}}{T} \int_0^T |\frac{dB}{dt}|^{\alpha} dt; \tag{10}$$

however, in this approach, the waveform of the excitation is not divided into minor and major loops. It is directly applied to the whole period. In [103], a similar, but simple approach was published with the name waveform coefficient Steinmetz equation. Other newer approaches such as the dual-natural Steinmetz equation use the combination of two iGSE methods, one to model the hysteresis and a second one to model the dynamic losses [112,113].

These Steinmetz-formula-based approaches offer a simple and fast way to predict iron loss without deep knowledge and without measurements of the materials. The weakness of these formulas is that the Steinmetz coefficients should vary with the applied frequency. This results in the accuracy of these formulas being lower than other mathematical-description-based methodologies, such as the Preisach or Jiles–Atherton method, especially at lower frequencies, where the losses mainly depend on the effect of the hysteresis. The rotational losses are also not considered by the previously mentioned formulas [106,109,111].

### 4.2. Separation of the Losses

The other type of function approximation models is based on the loss separation methodology. The first approach was introduced by Jordan in 1924 [114]. In this approach, the hysteresis and eddy-current-based losses (also called dynamic losses) are modelled separately. The model assumes that the hysteresis losses are caused by irreversible magnetisation effects (Brakhausen jumps) [115,116], while the dynamic losses are related to the generation of the eddy current near the domain walls:

$$P = P_{hyst} + P_d = C_h f B^2 + C_d f^2 B^2, \tag{11}$$

where $P_{hyst}$ means the hysteresis losses, $P_d$ represents the dynamic eddy current losses, while the $C_h$ and $C_e$ parameters can be determined from a measurement. The model expects that the hysteresis losses at low frequencies mainly cause the losses. The eddy current

losses can be calculated in a laminated thin sheet, and it can be written in the following form [109,117]:

$$P_{ec} = \frac{\sigma d^2}{12\rho} \left( \frac{dB(t)}{dt} \right)^2,$$ (12)

The eddy current loss term can be derived from Maxwell's equations, where $d$ is the lamination thickness, $\rho$ is the specific mass of the magnetic steel, and $\sigma$ represents the specific conductivity. The dynamic loss ($P_{dyn} > P_{ec}$) is found to be generally larger than the above-described, skin-effect-based description (Equation (12)). Pry and Bean [118] proposed to introduce the term for excess losses as a correction factor for dynamic loss. They considered an infinite lamination, which contains a lattice of periodic arrays of a *2L* long longitudinal domains. The correction factor can be writtenin the following term when $2L/d >> 1$:

$$P_{exc} \ (1.63\frac{2L}{d} - 1) \cdot P_{ec}.$$ (13)

However, this model was the first to show the need for excess losses. The formula's applicability is limited due to its highly ideal character [119]. In the case of microcrystalline materials, excess losses were found to be similar to the eddy current losses $P_{ec}$ $P_{exc}$ [120,121]. In the case of SiFe alloys, this formula is not be applicable to describe the anomalous losses, which shows a nonlinear dependence on the frequency [109,122,123], Bertotti [119,121] developed a statistics-based theory to calculate these anomalous losses with the following formulae:

$$C_{exc} = \sqrt{SV_0\sigma G},$$ (14)

where $\sigma$ is the electric conductivity, $S$ is the cross-sectional area of the lamination, $V_0$ is related to the grain size and the local forces, while $G$ represents a dimensionless coefficient [117,122,124]. The three-term iron loss formula can be written in the following form:

$$p = C_h f B^{alpha} + C_d f^2 B^2 + C_{exc} f^{1.5} B^{1.5}.$$ (15)

The coefficients can be obtained from a function fitting to the experimental data on different frequencies. This is the Bertotti model, which is widely used in the industry [116,119,125–128]. This and the modified Bertotti model, where the sum of the hysteresis losses at different frequencies is used to model the hysteresis loss of the material [129], is implemented in many commercial FEM tools (Flux [30,130,131]) [92,130,132,133].

Some recent studies have shown that more-accurate results can be achieved if the third anomalous loss coefficient is omitted, in the case of non-oriented steels [117,125,134,135]. These formulas can be written in the following form:

$$p = C_h(f, B)fB^\alpha + C_d(f, B)f^2B^2,$$ (16)

where the main difference is that the hysteresis loss ($C_h$) and the dynamical loss ($C_d$) coefficients depend on the measured frequency and the magnetic flux density, as well. In [135–137], the authors presented in their papers that in a certain frequency range, the values of the $C_d$ and $C_{exc}$ coefficients only depend on the flux density.

In recent papers [116,117,138], the authors proposed the temperature-dependent form of these equations, which is considered in the second, dynamical loss as the only temperature-dependent component of the formulae:

$$P = C_h f B^\alpha + \frac{C_d^2 B^2}{1 + \vartheta(T - T_0)} + C_{exc}^{1.5} B^{1.5},$$ (17)

where $\vartheta$ is an additional parameter identified by the measured temperature-dependent loss data and $T$ is the temperature in degrees Celsius [138]. It is supposed that, in the examined case, the excess loss term does not depend significantly on the temperature.

Rotational Losses

In electrical machines, there is another important factor that should be considered: these machines create a rotating electrical field, which has a time-dependent character. This phenomenon by which the hysteresis has some lagging effect in a rotating machine was first described by [139]. Kaplan [140] made measurements on highly grain-oriented and non-grain-oriented steels, and he found that these lower-loss, high-grain-oriented steels have higher losses under rotating conditions. Several papers were published to describe the measurement of SMC materials and the numerical analysis of these materials in rotating machines [141–148].

A separation-based approach was introduced to describe the rotating losses [149]:

$$P_r = P_{hr} + P_{er} + P_{ar} \tag{18}$$

where $P_r$ means the total amount of the rotating losses, $P_{er}$ represents the eddy current losses, while $P_{ar}$ are the anomalous losses. An empirical, curve-fitting approach can describe the anomalous losses in the following form [149]:

$$P_{ar} = C_{ar}(fB)^{1.5}, \tag{19}$$

where $C_{ar}$ is an empirical coefficient to describe the losses, $f$ is the magnetisation frequency, and $B$ is the magnitude of the applied flux density.

In rotating machines, the flux density can be considered elliptically rotating with harmonics [147,150]. These flux-given densities have maximum and minimum values for every $k$th harmonic excitation. Their ratio can be calculated and denoted by $R_{Bk} = B_{kmin}/B_{kmax}$ for the $k$th harmonic. Using this formula, the rotating losses can be calculated in the following way:

$$P_t = \sum_{e=1}^{N_e} \sum_{k=1}^{\infty} (P_{rk}R_{Bk} + (1 - R_{Bk})^2 P_{ak}) \tag{20}$$

where $N_e$ is the number of elements of the core material, $R_{Bk}$ is the scaling factor from the axis ratio of the $k$th harmonic ellipse, $P_{ak}$ is the alternating loss with flux density, $B_{kmaj}$, and $P_{rk}$ is the purely rotational loss.

*4.3. Advanced Mathematical Models for the Hysteresis*

Hysteresis characteristics' modelling is also suitable for describing the iron core material and modelling the losses. Using hysteresis characteristics, the loss can be determined by the solution of the following equation:

$$P = \int_0^T \left( H(t) \frac{dB(t)}{dt} \right) dt, \tag{21}$$

where $H(t)$ and $B(t)$ are the hysteresis model's input magnetic field and output magnetic flux density if the model is forward, also called the direct model. In the case of the inverse model, $B(t)$ is the input and $H(t)$ is the output.

There are mainly three physical processes that cause the hysteresis effect in magnetic materials [151]: domain wall motions [151–154] and the nucleation of domain walls [151]. Many different models have been proposed to explain the hysteresis behaviour in materials with small and large grain sizes. A large grain size means the domain wall thickness is above the single domain particle size [155].

Several approaches have been published in the literature to describe the mathematical relation for the B–H curve. These methodologies can be categorised in many ways. Mörée showed that most of them can be mathematically described as Play- and Preisach-type models [151]. The most-general model is the Preisach model because it allows the inclusion of saturation and variations of the hysteresis [151]. Berqgvist proposed a friction-like magnetic hysteresis model, a proposition of the Jiles–Atherton and Preisach models [156,157].

Jiles and Atherton proposed a physical-based description of magnetic hysteresis in [158]. This model uses the Langevin function to describe the anhysteretic curve and a differential equation to describe the hysteresis behaviour [141,159–161]. Identifying the parameters for the Jiles–Atherton model parameters is a complex task, like the Preisach model parameters. In the following subsections, we will show the classical and the dynamical Preisach model. Moreover, we will show the application of the classical and dynamic Preisach model on a simple lamination.

Classical Preisach-Model

One of the most-commonly used methods to approximate the hysteresis characteristic is the Preisach model [162–165]. According to this model, the resulting hysteresis characteristic is given by the weighted sum of the outputs of an infinite number of elementary hysteresis terms, defined by the following formula [162]:

$$B(t) = \mathcal{H}\{H\} = \iint_{\alpha \geq \beta} \mu(\alpha, \beta)\hat{\gamma}(\alpha, \beta)H(t)d\alpha d\beta. \tag{22}$$

The characteristics of hysteresis are shown in Figure 10 with an up value of $\alpha$, a down value of $\beta$, and $\alpha \geq \beta$; its output takes only two values, so $\hat{\gamma}(\alpha, \beta)H = \pm 1$. $\mu(\alpha, \beta)$ is the Preisach distribution function, which weights the output of each hysteresis. A block diagram representation of the Equation (22) is shown in Figure 10, where the parameters $\alpha$ and $\beta$ of each hysteresistakes different values in each parallel branch.

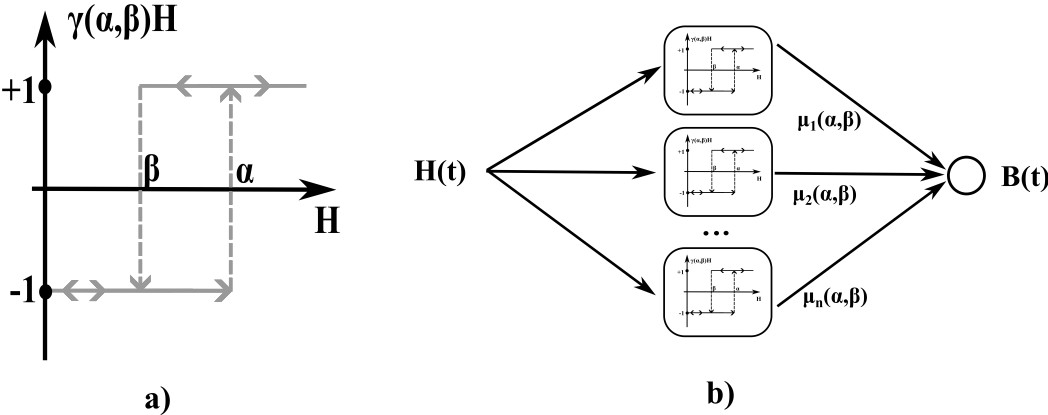

a)                                        b)

**Figure 10.** (**a**) shows the characteristics of a single hysteresis, while (**b**) shows the complete hysteresis model, which is built from the sum of these hystereses.

To describe this, the Preisach triangle was developed [30,162] and is shown in Figure 11. The Preisach triangle is the part of the plane $\alpha - \beta$ for which it is true that $\alpha \geq \beta$. The prior life of the system is defined by the staircase line $L(t)$, which moves from left to right as the input signal increases and from top to bottom as the input signal decreases and couples the hysteresis terms $\alpha$ and $\beta$ with its movement. In the default position, the stepped curve connects the points $(0; 0)$ and $(+1; -1)$, thus bisecting the triangle. The corners of the staircase line are formed by the minimum and maximum values of the input signal, thus acting as a memory of the system.

Several approaches are published in the literature to resolve the double-integral in (22). Szabó decomposed the function in the dual-integral, which represents $\mu$, into a product of univariate Gaussian functions. In this way, the double-integral was reduced to a closed, analytic form [166,167]. Fuzi in [168] extended the Preisach distribution functions to approximate the main hysteresis loop better. Moreover, he introduced non-congruent minor behaviour. Because the original description of the Preisach model has two important mathematical properties, it has congruent minor loops and non-accommodating minor loops [151,160,169].

The Everett function [160,169] can be used efficiently instead of the distribution function, resulting in a model without the double-integral (22), i.e., the implemented model can be numerically more efficient than the previous approaches:

$$E(\alpha, \beta) = \iint_{\alpha \geq \beta} \mu(\xi, \eta) \mathrm{d}\xi \mathrm{d}\eta. \tag{23}$$

The model output can then be obtained by

$$B(t) = -E(\alpha_0, \beta_0) + 2 \sum_{k=1}^{K} [E(\alpha_k, \beta_{k-1}) - E(\alpha_k, \beta_k)], \tag{24}$$

where $K$ denotes the number of stairs in $L(t)$.

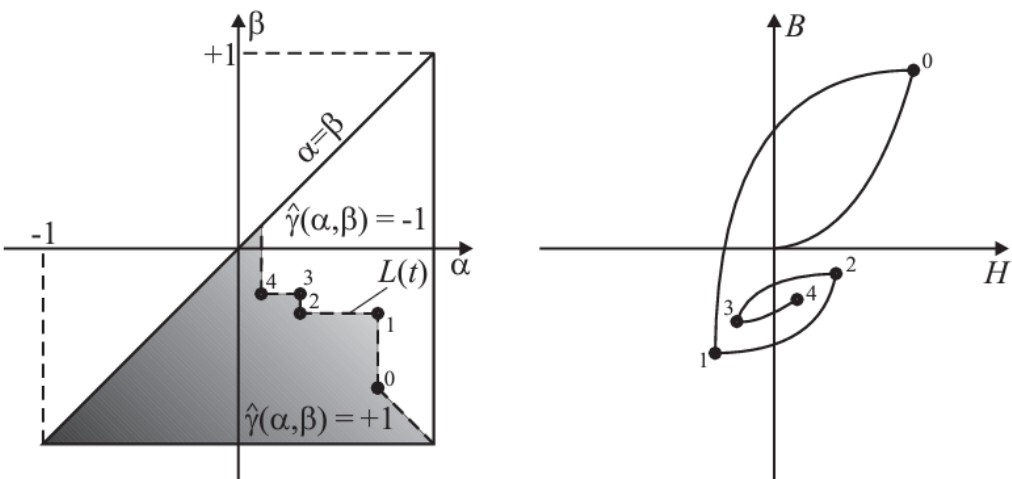

**Figure 11.** The Preisach triangle with the staircase line and the corresponding hysteresis loop [162].

The other advantage of the Everett function is the direct connection with the measured first-order reversal curves or the concentric loops. The application of this methodology needs to resolve a computationally demanding parameter-determination process on the measurement results [170–174].

*4.4. Dynamic Preisach Model*

Due to the recent advances in electrical drives and electrical machine design, it was necessary to consider the frequency dependency of the losses more accurately during the machine design process [175,176].

The classical Preisach model does not consider frequency dependency. It only models the magnetic flux density as a function of the magnetic field intensity, i.e., $B = \mathcal{H}\{H\}$. Its inverse obtains the magnetic field intensity as a function of the magnetic flux density, i.e., $H = \mathcal{B}\{B\}$ [30,177,178]. Some authors refer to this dynamic Preisach model as the inverse Preisach model [164,179,180].

Magnetic field intensity is derived as the superposition of two terms: $H = H_{static} + H_{dynamic}$, where $H_{static}$ is given by the inverse static model and $H_{dynamic}$ is the magnetic field intensity due to dynamic effects. The approach in [175] builds up a surface $S(B, dB/dt)$, which is a function of two variables, the magnetic flux density and its variation. The Preisach distribution function can also be extended to consider the frequency dependency, i.e., applying $\mu(\alpha, \beta, f)$ [160,169]. These models are considered separately as loss surface models in the literature [30,175].

Hysteresis of a Lamination

The previous parts of this section showed how to model iron losses by simple formulas based on physical considerations. It is shown that standardised measurements can identify model parameters; afterwards, the models can approximate the losses via FEM-based simulations. The Steinmetz equation or the loss-separation-based formulas are used in the post-processing stage of finite-element simulations. Therefore, during the calculations, these methods cannot accurately consider the nonlinear B–H characteristics of every calculated finite-element. This section highlights how the complex Preisach model can be applied in FEM simulations to obtain the losses inside such a simple arrangement, i.e., inside a lamination made of material M250-35A. The simple illustration shows that applying the Preisach model results in accurate loss calculation; however, if the anomalous losses term extends it, it results in a time-consuming algorithm. In this case, obtaining a convergent algorithm can be difficult for a more-complex geometry. However, the calculated hysteresis loops at every finite-element cell of the arrangement can be more exact, which can benefit custom-designed electrical machines from SMLC materials. Two kinds of Preisach-type models are applied in this section for a simple lamination to estimate the losses of the selected M250-35A-grade steel and demonstrate the applicability of the Preisach models.

The quasi-static Maxwell's equations can approximate electromagnetic fields inside a lamination [162,181]:

$$\nabla \times \boldsymbol{H} = \sigma \boldsymbol{E},$$
$$\nabla \times \boldsymbol{E} = -\frac{\partial \boldsymbol{B}}{\partial t}, \tag{25}$$
$$\nabla \cdot \boldsymbol{B} = 0,$$

where $\boldsymbol{H}$ is the magnetic field intensity, $\boldsymbol{E}$ is the electric field intensity, $\sigma \boldsymbol{E}$ is the eddy current with the conductivity denoted by $\sigma$, and $\boldsymbol{B}$ is the magnetic flux density.

A one-dimensional finite-element-method-based code was implemented in Matlab to approximate the electromagnetic fields and the losses inside the lamination. Maxwell's equations have the following simpler formula in 1D:

$$\frac{\partial H_y}{\partial x} = \sigma E_z,$$
$$\frac{\partial E_z}{\partial x} = \frac{\partial B_y}{\partial t}. \tag{26}$$

The equation $\nabla \cdot \boldsymbol{B} = 0$ is satisfied automatically. It is supposed that the magnetic field intensity and the magnetic flux density have only components in the $y$ axis ($H_y$ and $B_y$), and the electric field intensity only varies along the $z$ axis ($E_z$). The lamination depth is in the $x$ axis [162,181].

The constitutive relationship between the magnetic field intensity and the magnetic flux density is given by the polarisation formulation [162,182], i.e.,

$$B_y = \mu(H_y - H_{y,\text{exc}}) + R_y. \tag{27}$$

The term $H_y - H_{y,\text{exc}}$ is the input magnetic field intensity of the Preisach model and $\mu$ is constant:

$$\mu = \frac{\mu_{\max} + \mu_{\min}}{2}, \tag{28}$$

where $\mu_{\max}$ and $\mu_{\min}$ are the characteristics' maximum and minimum slope [182]. The excess field term can be expressed according to the viscous-type model as follows [183]:

$$H_{y,\text{exc}} = \delta \left| \frac{1}{r(B_y)} \frac{\mathrm{d}B_y}{\mathrm{d}t} \right|^{1/\gamma}, \quad \text{and} \quad r(B_y) = \frac{R_0}{1 - \left(\frac{B_y}{B_s}\right)^2}, \tag{29}$$

where $\delta$ is positive or negative along an increasing or decreasing curve of the hysteresis loop, respectively, and $\gamma$ and $r(B_y)$ measure the dynamic loops. The fixed-point iteration scheme gives the residual term $R_y$. The lamination is made of the ferromagnetic material M250-35A with $R_0 = 23.79$ and $\gamma = 1$.

After a short manipulation, the following partial differential equation can be obtained:

$$-\frac{\partial^2 H_y}{\partial x^2} + \mu\sigma\frac{\partial H_y}{\partial t} = \mu\sigma\frac{\partial H_{y,\text{exc}}}{\partial t} - \sigma\frac{\partial R_y}{\partial t}. \tag{30}$$

This nonlinear partial differential equation is solved by the finite-element method at every time instant of the source magnetic field defined as a sinusoid. The fixed-point algorithm of a time step to solve the nonlinear problem is as follows [162,182]:

- Solve (30) by the finite-element method (start with the value of $R_y$ and $H_{y,exc}$ of the previous time instant), which gives $H_y$ at every node of the mesh;
- The magnetic flux density $B_y$ at every node can be obtained by the Preisach model with the input of $H_y - H_{y,exc}$;
- The residual term $R_y$ is given by rearranging (27), i.e., $R_y = B_y - \mu(H_y - H_{y,\text{exc}})$;
- The term $H_{y,exc}$ is updated by (29).

These steps are repeated until convergence.

The total loss $P$ can be obtained by

$$P = \frac{1}{T\rho}\int_0^T H_y\frac{\mathrm{d}B_y}{\mathrm{d}t}\mathrm{d}t, \tag{31}$$

where $H_y$ and $B_y$ are the magnetic field intensity at the finite-element node of the lamination surface and the magnetic flux density averaged over the lamination, the mass density of the lamination material is denoted by $\rho$, $\rho = 7600\,\text{kg/m}^3$, and $T$ represents the examined time period. The unit of $P$ is Watts per kilogram.

Table 1 presents a comparison between the simulated iron losses and the iron loss data obtained from the measured curves performed by our hysteresis loop measurement system [117]. The measured data are given in parentheses. The above-mentioned finite-element-method-based eddy current field simulation gives simulated iron loss data. The measured hysteresis loops were used to identify the classical and the extended Preisach model. Two simulated loss data are given in Table 1, calculated without and with the excess loss term $H_{y,\text{exc}}$. The results obtained by the static Preisach model (when $H_{y,\text{exc}} = 0$) take only eddy currents into account, and excess losses are neglected, while excess losses can be taken into account by applying the term $H_{y,\text{exc}}$.

Some simulated hysteresis loops are depicted in Figure 12, where the effect of eddy currents, as well as excess losses can be seen, i.e., the hysteresis loop is increased by the increased frequency. Some comparisons between the measured and simulated data can be studied in Figure 13, where concentric minor loops and higher-order minor loops are generated and simulated.

**Table 1.** Comparison of simulated (eddy current only/extended) and measured losses.

| B [T] | P @ 5 Hz | P @ 50 Hz | P @ 100 Hz | P @ 200 Hz |
|-------|----------|-----------|------------|------------|
| 0.2 | 0.01/0.01 (0.01) | 0.08/0.093 (0.08) | 0.17/0.22 (0.21) | 0.39/0.55 (0.55) |
| 0.6 | 0.05/0.05 (0.05) | 0.54/0.616 (0.60) | 1.16/1.49 (1.52) | 2.67/3.98 (4.10) |
| 1.0 | 0.12/0.12 (0.12) | 1.30/1.50 (1.38) | 2.85/3.66 (3.74) | 6.69/9.96 (9.96) |
| 1.4 | 0.23/0.23 (0.22) | 2.53/2.85 (2.57) | 5.54/6.87 (6.78) | 13.1/18.4 (17.7) |

Calculating with a function implementing the Preisach model accounts for about 80–90% of the total finite-element-method-based calculation. The generation of the required values of the Everett function within the Preisach model is about 95–97% of the total model run.

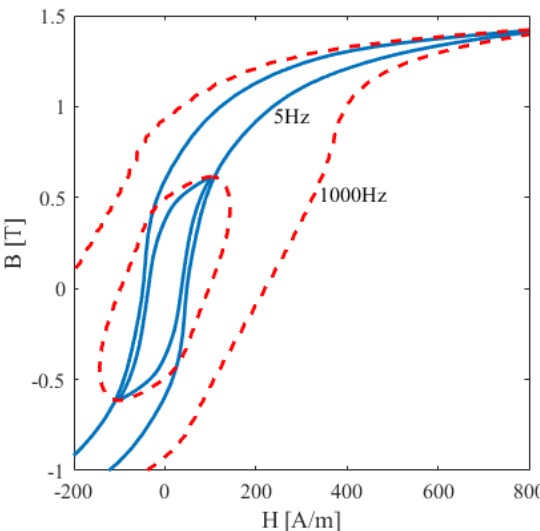

**Figure 12.** Simulated hysteresis loops at 5 Hz and 1000 Hz, 0.6 T and 1.4 T, simulated by 1D FEM lamination model.

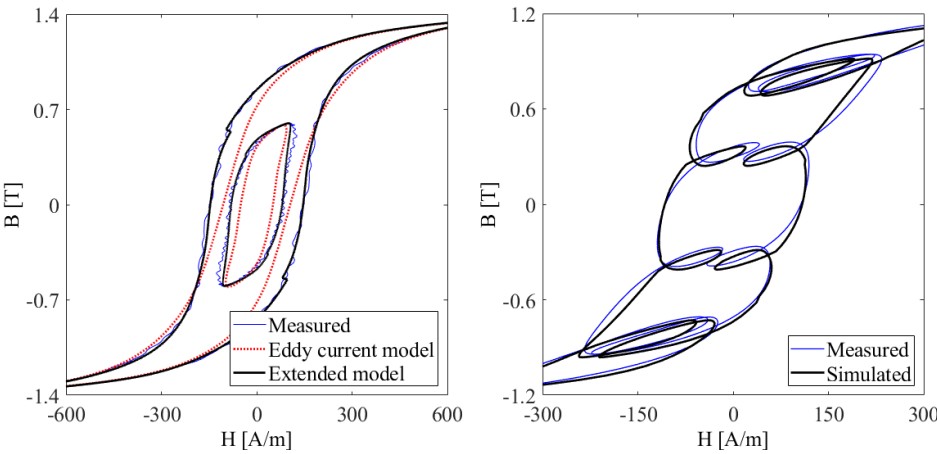

**Figure 13.** Measured and simulated dynamic hysteresis loops.

## 5. Conclusions

Three-dimensional printing is a promising technology for creating iron cores for electrical machines. However, the applied materials are still expensive. It can be an economical technology for creating rapid prototypes of iron cores or custom-manufactured machines with complex geometries. One possible material is the Fe-6.5wt%Si silicon steel alloy, which can reduce the operating and operation costs. However, it is incredibly brittle due to its high silicon content. An exciting possibility with this technology is that the formation of the magnetic domain structure can be directed by applying an appropriate magnetic field during the printing process. FeSi alloys are promising materials for creating competitive electrical machine designs in the future. However, improving the ductility of this material is not enough to harness the full potential of this material. A better understanding of the magnetisation process and a deep knowledge of the numerical iron loss calculation methods are necessary to create more-competitive electrical machine designs. The second part of the paper overviewed the applicable measurement methodologies and iron loss calculation methods. Most of the proposed methodologies can be used to calculate the losses in 3D-printed materials. However, the Preisach- or Play-model-based calculations seem to be the most-promising iron loss models. Besides their high accuracy, the application of these models requires an extensive measurement methodology; the direct application of this methodology on 3D-printed prototypes is not straightforward, but a challenging problem.

**Author Contributions:** Conceptualization, T.O., writing—original draft preparation, T.H., B.T., M.K., T.O. and B.K.; writing—review and editing, T.O., B.K. and M.K.; visualization, T.H., B.K., M.K. and T.O.; supervision, T.O. and M.K.; project administration, T.O.; funding acquisition, T.O. All authors have read and agreed to the published version of the manuscript.

**Funding:** This research received no external funding.

**Data Availability Statement:** Not applicable.

**Conflicts of Interest:** The authors declare no conflict of interest.

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
