# Peer review of "Iron Loss Calculation Methods for Numerical Analysis of 3D-Printed Rotating Machines: A Review"

_energies, doi:10.3390/en16186547_

Round 1

Reviewer 1 Report

Congratulations for your work.

Author Response

Dear Reviewer,

 Thank you for your positive feedback.

Reviewer 2 Report

The review needs a reworked summary to clarify the findings of the paper. A discussion paragraph should be added to chapter 4 in order to compare the presented models. It seems that the review includes research results of the authors (figures 4-8, 14-16), these should be removed to a separate research paper or referenced properly. The use of the term soft magnetic composites (SMCs) should be changed to something more appropriate for the benefit of the reader.

Comments:

* The usage of the term “soft magnetic composites (SMCs)” throughout the text can be misinterpreted by the reader. SMCs usually denote “materials which are made of iron powder particles coated with an electrically insulating layer and can be formed into complex shapes by means of powder metallurgy.” If I understand correctly, the authors wish to discuss materials which are a combination of two materials with different physical and chemical properties (of which one is magnetic and both are 3D printed). Your manuscript needs some other acronym (not SMC – it is already well known as something entirely different) and a lengthy explanation in the introduction what you wish to express.

*On the same topic, a few comments. First, in the beginning of the introduction: “The additive manufacturing of Soft Magnetic Composites (SMC) is an increasingly important area.”  I believe no one has actually published any meaningful results of 3D printed SMCs – from both the definition of a multi-material magnetic core and from the definition of the coated insulated particles. If I am mistaken add references. Perhaps additive manufacturing of soft magnetic materials is an increasingly important area.

*Line 86 – excessive []

*Figure 3 – add an explanation to all of the properties discussed and explain in more detail the categorization.

*Line 144 – “In the Powder - DED and EBM processes, it is relatively easy to alternate metallic and ceramic materials by employing a suitable adjustment of the powder dosage and the applied energy density.” – add references of joined multi material samples!

*Figures in the manuscript which refer to other works must have references. Figures 4-8? If these are your own research results, they don’t belong to a review paper!

*Chapter 3 “Measurement methodologies for iron losses” begins with microstructural analysis. It should begin with iron losses and measurement methodologies. Microstructural information should be moved to a separate chapter or to the end of the chapter 3.

A discussion / comparison / analysis of the different iron loss models must added to the manuscript for the benefit of the reader

*4.4.1. Hysteresis of a lamination – figure 14, and figure 15 – are these unpublished research results? They belong to a separate research paper!

*Figure 16 – there should be no Figures in the Conclusions chapter!

*The conclusion should be rewritten to capitalize on the key information of this review – it is not specific enough. Line 575 – “The first part of the paper has shown a current overview of the 3D printing technology and its possibilities for iron loss reduction, while the second part’s central role is to overview the applicable measurement methodologies and iron loss calculation methods” – bring out the most important findings of the review.

The language use is fine

Author Response

Dear Reviewer,

 The authors would like to thank the time Reviewer for providing comments. We hope that our answers and the changes in the manuscript clarify all the questions and enhance the quality and value of the manuscript.
An effort has been made to address all the concerns of the reviewers and to
accommodate all their suggestions to enrich the revised manuscript. All our answers and the changes are highlighted by blue color.

Comments:
The usage of the term “soft magnetic composites (SMCs)” throughout the text can be misinterpreted by the reader. SMCs usually denote “materials which are made of iron powder particles coated with an electrically insulating layer and can be formed into complex shapes by means of powder metallurgy.” If I understand correctly, the authors wish to discuss materials that combine two materials with different physical and chemical properties (of which one is magnetic, and both are 3D printed). Your manuscript needs some other acronym (not SMC – it is already well known as something entirely different) and a lengthy explanation in the introduction of what you wish to express.

Thank you for your comment. The term composite also covers 3D printed
layered multimaterial structures, but we agree that the term may misinterpreted by some readers. In order to remedy the problem, we introduce the Soft Magnetic Layered Composite (SMLC). In this regard, we have made the following modifications (red indicates deleted content and blue indicates new content):

The additive manufacturing of Soft Magnetic Materials (SMM) is an increasingly important area [1-5]. Within this, the design of layered composite structures made from multiple materials is given rise to a new research field that draws parallels between laminated steel sheet cores and 3D-printed layered structures [6-9]. In the following sections, we will refer to these materials as Soft Magnetic Layered Composites (SMLCs). The term layered structure typically refers to an alternating arrangement of metal and electrically insulating materials. We usually discuss two-component systems, but the simultaneous use of three or more material can also occur.

* On the same topic, a few comments. First, in the beginning of the introduction: “The additive manufacturing of Soft Magnetic Composites (SMC) is an increasingly important area.” I believe no one has actually published any meaningful results of 3D printed SMCs – from both the definition of a multi-material magnetic core and from the definition of the coated insulated particles. If I am mistaken add references. Perhaps additive manufacturing of soft magnetic materials is an increasingly important area.

Thank you for your comment. The term of SMC was modified in the manuscript.

Line 86 – excessive []

Thank you for your remark,  the missing citations inserted to the text.

Figure 3 – add an explanation to all of the properties discussed and explain in more detail the categorization.

Thank you for your comment. We have made the following changes:

The coercive field of soft magnets increases with decreasing grain size. The
grain size dependence of coercivity is explained by a size-dependent demagnetization factor. The phenomenological description was given by Mager [46] in the form of a linear relation between coercive field and the reciprocal of grain size (1/GS), in which the slope is proportional to the ratio of domain wall surface energy (γ) to saturation polarization (Js). The effect of grain size is evident for the AM SMLC and powder cores. For layered structures, the detrimental effect of surface pining should be taken into account. Considering the effectiveness of these effects, we have prepared the soft magnetic parameters (Hc, μstatic, and flim) evaluation table presented in Figure 3. The limitation of the cut-off frequency is determined by the dominance of eddy currents. By reducing the layer thickness and preventing the formation of a large current path, it is possible to extend the operational frequency range.

*Line 144 – “In the Powder - DED and EBM processes, it is relatively easy to alternate metallic and ceramic materials by employing a suitable adjustment of the powder dosage and the applied energy density.” – add references of joined multi material samples!

Thank you for your comment, we inserted some relevant references.

Figures in the manuscript which refer to other works must have references. Figures 4-8? If these are your own research results, they don’t belong to a review paper!

Thank you for your comment, we added the missing references and replaced the pictures according to your suggestions. Figure 8 was removed from the paper.

Changes in the text:

From line 230 to 236

Oliveira et al. investigated the build orientation effect on magnetic properties of managing steel 300. They conclude that there is likely a correlation between the residual stresses and coercivity, remanence. The ring-like specimens printing orientation were XYZ, XYZ-45° and YZX. Figure 4 shows the magnetization curves and hysteresis loops of these samples.
M. Garibaldi et al. examined the grain orientation and structure for different print orientations which are illustrated in Figure 5 for Fe-6.5wt \%Si composition.

From line 245 to 247:

M. Garibaldi et al. investigated the effect of laser printing parameters on the microstructure and crack formation. It has been shown that irregularly shaped porosities typically form at the interfaces of shallow and wide melt pools, while spherical pores form at the bottom of deeper melt pools. They also proved that the cumulative crack length (c.c.l.) parameter starts to increase significantly above an energy input of 280 J/m. However, the elimination of large, irregularly shaped pores can also be eliminated from the microstructure in the case of an energy input of 280 J/m.} These results are shown in Figure 6.

*Chapter 3 “Measurement methodologies for iron losses” begins with microstructural analysis. It should begin with iron losses and measurement methodologies. Microstructural information should be moved to a separate chapter or to the end of the chapter 3.

 Thank you for your comment. The description of the microstructure is necessary when discussing the measurement methods because the choice of the method depends, among other things, on the microstructure and the applied manufacturing technology. We would like to draw the reader's attention to this important fact.

Changes in the text:

When measuring the magnetic properties of 3D printed parts, it is important to remember that the manufacturing technology, the geometry of the printed part, its location in the printing space and its material composition all influence the selection of the appropriate measurement method.

A discussion / comparison / analysis of the different iron loss models must added to the manuscript for the benefit of the reader

The 4.th chapter contains this comparison of two kind of Preisach model. A small paragraph added to the beginning of this chapter to highlight the goal of this chapter.

Changes in the text:

The previous parts of this section showed how to model iron losses by simple formulas based on physical considerations. It is shown that standardized measurements can identify model parameters; afterwards, the models can approximate losses via FEM-based simulations. In this section, we would like to highlight how the  Preisach model can be applied in FEM simulations to get the losses inside a simple arrangement, i.e. inside a lamination made of material M250-35A. Two kinds of Preisach models are applied to estimate the losses of the selected steel.

*4.4.1. Hysteresis of a lamination – figure 14, and figure 15 – are these unpublished research results? They belong to a separate research paper!

Figure 14 removed from the paper.  The goal of this section is to show, how can 2 Preisach type model can be applied for the loss estimation of a selected magnetic steel. Figure 14 removed and Figure 15 shows the results of the calculation.

*Figure 16 – there should be no Figures in the Conclusions chapter!

The figure removed from the Conclusions chapter.

*The conclusion should be rewritten to capitalize on the key information of this review – it is not specific enough. Line 575 – “The first part of the paper has shown a current overview of the 3D printing technology and its possibilities for iron loss reduction, while the second part’s central role is to overview the applicable measurement methodologies and iron loss calculation methods” – bring out the most important findings of the review.

The conclusion section changed to highlight the findings of the literature survey:

3D printing is a promising technology for creating iron cores for electrical machines. However, the applied materials are still expensive.
It can be an economical technology for creating iron cores rapid prototypes or custom-manufactured machines with complex geometries. One capable material is the Fe-6.5wt\%Si silicon steel alloy, which can reduce operating and operation costs.
However, it is incredibly brittle due to its high silicon content. An exciting possibility with this technology is that the formation of the magnetic domain structure can be directed by applying an appropriate magnetic field during the printing process. 
FeSi compounds are promising materials for creating competitive electrical machine designs in the future. 
However, improving the hardness of this material is not enough to harness the full potential of this material.  A better understanding of the magnetization process and a deep knowledge of the numerical iron loss calculation methods are necessary to create more competitive electrical machine designs.
The second part of the paper overviewed the applicable measurement methodologies and iron loss calculation methods. Most of the proposed methodologies can be used to calculate the losses in 3D printed materials.  However, the Preisach or play model-based calculations seem the most promising iron loss models.  Besides their high accuracy, the application of these models needs an extensive measurement methodology, the direct application of this methodology on 3D printed prototypes is not straightforward, a challenging problem.

Reviewer 3 Report

The article is a review and concerns a new technique for creating cores of electrical machines. However, the discussed technology is very laborious and expensive, and its effects are not fully demonstrated by the authors. It is a pity that the obtained magnetizability and loss characteristics were not compared with the characteristics of materials commonly used for the cores of electric machines. As shown in Fig. 15, the strength of the core material proposed by the authors at a frequency of 1000 Hz and induction 1.4T is about 150 W/kg. Similar results can be obtained using, for example, M270-35A electrical sheet metal with a thickness of 0.35 mm. Much better results are obtained with the use of NO20 sheet (with a wear capacity of 74W/kg at 1.4T induction and 1000Hz frequency), while the commonly used mechanical punching technology can be used to make cores from these sheets. Much better results are obtained for cores made of amorphous sheet, whose wear resistance for the given conditions is of the order of 5 W/kg. I believe that the article should be extended in this respect.

Author Response

Dear Reviewer,

The authors would like to thank the reviewers for the time they spent to provide comments. We hope that our answers and the changes in the manuscript clarify all the questions and enhance the quality and value of the manuscript. An effort has been made to address all the concerns by the reviewers and to accommodate all their suggestions to enrich the revised manuscript.

We added a small comparison from the BH characteristics of one current magnetic steel (M270) and some other 3D printed materials. You are absolutely right; the amorphous materials have much better iron loss characteristics than the proposed materials.  However, the paper's goal is not to propose the superiority of these materials over conventional magnetic steels or amorphous alloys. This 3D printing technology makes it possible in the future to create prototypes with competitive parameters with mass-manufactured electrical machines for a reliable price.

The paper aimed to show that it is crucial to understand and improve the current iron loss models to accurately calculate iron losses in these prototypes.

We  modified the part of the conclusions section to highlight this better:

3D printing is a promising technology for creating iron cores for electrical machines. However, the applied materials are still expensive.
It can be an economical technology for creating iron cores rapid prototypes or custom-manufactured machines with complex geometries. One capable material is the Fe-6.5wt\%Si silicon steel alloy, which can reduce operating and operation costs. However, it is incredibly brittle due to its high silicon content. An exciting possibility with this technology is that the formation of the magnetic domain structure can be directed by applying an appropriate magnetic field during the printing process.  FeSi compounds are promising materials for creating competitive electrical machine designs in the future.  However, improving the hardness of this material is not enough to harness the full potential of this material.  A better understanding of the magnetization process and a deep knowledge of the numerical iron loss calculation methods are necessary to create more competitive electrical machine designs.
The second part of the paper overviewed the applicable measurement methodologies and iron loss calculation methods. Most of the proposed methodologies can be used to calculate the losses in 3D printed materials.  However, the Preisach or play model-based calculations seem the most promising iron loss models.  Besides their high accuracy, the application of these models needs an extensive measurement methodology. The direct application of this methodology on 3D printed prototypes is not straightforward, a challenging problem.

Round 2

Reviewer 2 Report

The manuscript is still quite rough and needs some additional work. Comments:

*Add references to each hysteresis loop on Figure 1. Note the magnetization frequency of the loops. “The measured 3D printed composites are made from FeCo, FeNi, and FeSi composites” – FeCo, FeNi and FeSi are not “composites”. – this must be fixed.

*flim is still not explained for Figure 3.

* Please go over the text again, there are several instances of strange sentences and misleading information:

”There are several kinds of research on the printability of Fe-6.5wt%Si with zero magnetostriction, which has cheap but relatively good soft magnetic properties [22,36,74,75]” (227) – cheap magnetic properties?

“A solution to avoid structural defects and high brittleness is the development of a gradient composition, which can be obtained by appropriate layering and laser sintering of elemental iron and silicon powders.” (Line 238) – the layered structure in Figure 7 is not gradient in nature

 “However, improving the hardness of this material is not enough to harness the full potential of this material.” (Line 612) - The problem of Fe6.5Si is not "hardness" but "ductility"

“The 3D printability of Fe35Co65 or the extremely soft Fe20Ni80 alloys with the highest known saturation magnetization, known under the Permalloy brand name, is a relatively easy task since standard steel powders often contain significant amounts of Ni and Co in addition to Fe.” (Line 215)   - what is this highest known saturation magnetization? Permalloy saturates at roughly around 1 T. Also, I wouldn’t say that pure Fe is difficult to print and thus the “significant amounts of Ni and Co” don’t really improve the printability of iron; but rather some specific alloys are difficult to print (like Fe6.5Si).

”The printability of Fe-Si alloys becomes problematic somewhere around 3wt% silicon” (Line 218) – 3% content is standard silicon content for electrical steels and it is definitely cold-workable and printable – I can say that from experience. High silicon steel is problematic, 6.5% is far more brittle than 3%.

* Line 618 “However, the Preisach or play model-based calculations seem the most promising iron loss models” – analyse the methods in the end of chapter 4, provide the reader with some kind of a comparison and then based on that, make the conclusions and state which models are the best suited or promising for 3D printed materials. Then chapter 4.4.1 would make more sense – to show the reader how to best implement the identified method.

-

Author Response

Dear Reviewer,

The authors would like to thank the reviewers for the time they spent to provide comments. We hope that our answers and the changes in the manuscript clarify all the questions and enhance the quality and value of the manuscript. An effort has been made to address all the concerns of the reviewer and to accommodate all his suggestions in order to enrich the revised manuscript.

Point-by-point answers to the reviewer's questions, all of the required changes highlighted by red in the manuscript:

The manuscript is still quite rough and needs some additional work. Comments:

  • Add references to each hysteresis loop on Figure 1. Note the magnetization frequency of the loops. “The measured 3D printed composites are made from FeCo, FeNi, and FeSi composites” – FeCo, FeNi and FeSi are not “composites”. – this must be fixed.

Thank you for your comment. We have corrected this mistake and added the required references to Figure 1.

  • flim is still not explained for Figure 3.

Thank you for your comment. We supplemented the description and provided the definition of the cut-off frequency, supporting it with a reference.

From line 130:

"In addition to the internal coercivity, the static permeability and cut-off frequency values can qualify the softness of core-shell and laminated structures during measurements. \textcolor{red}{The cut-off frequency can be calculated from Snoek's law [46]. The cut-off frequency is where the real part of the complex permeability changes significantly. Figure 3 summarises their characteristic values for the two types."

  • Please go over the text again, there are several instances of strange sentences and misleading information:

”There are several kinds of research on the printability of Fe-6.5wt\%Si with zero magnetostriction, which has cheap but relatively good soft magnetic properties [22,36,74,75]” (227) – cheap magnetic properties?

Thank you for your comment. It is a misunderstanding because of the not proper language use, the corrected text:

Line 227:

"There are several kinds of research on the printability of Fe-6.5wt\%Si with zero magnetostriction, which is a comparatively cheap  material but has relatively good soft magnetic properties [22,36,75,76]."

  • “A solution to avoid structural defects and high brittleness is the development of a gradient composition, which can be obtained by appropriate layering and laser sintering of elemental iron and silicon powders.” (Line 238) – the layered structure in Figure 7 is not gradient in nature

Thank you for your comment. It is a gradient alloy, even if the microstructure does not represent this. It was shown in reference nr. 77. This reference supports this with a detailed compositional analysis.

Line 242:

The cross-sectional gradient changes of the elemental composition are shown in [77].

“However, improving the hardness of this material is not enough to harness the full potential of this material.” (Line 612) - The problem of Fe6.5Si is not "hardness" but "ductility"

Thank you for your comment. We have corrected this mistake.

Line 613:

However, improving the \textcolor{red}{ductility} of this material is not enough to harness the full potential of this material.

“The 3D printability of Fe35Co65 or the extremely soft Fe20Ni80 alloys with the highest known saturation magnetization, known under the Permalloy brand name, is a relatively easy task since standard steel powders often contain significant amounts of Ni and Co in addition to Fe.” (Line 215)   - what is this highest known saturation magnetization? Permalloy saturates at roughly around 1 T. Also, I wouldn’t say that pure Fe is difficult to print and thus the “significant amounts of Ni and Co” don’t really improve the printability of iron; but rather some specific alloys are difficult to print (like Fe6.5Si).

Thank you for your comment. We have corrected some mistakes in this section.

Line 216:

The 3D printability of high saturation magnetization Fe\textsubscript{35}Co\textsubscript{65} or the extremely soft Fe\textsubscript{20}Ni\textsubscript{80} alloys, known under the Permalloy brand name, is a relatively easy task since standard steel powders often contain significant amounts of Co and Ni in addition to Fe [63,70-72]. The printability of Fe-Si alloys becomes problematic somewhere around 6.5wt\% silicon [73].

”The printability of Fe-Si alloys becomes problematic somewhere around 3wt\% silicon” (Line 218) – 3\% content is standard silicon content for electrical steels and it is definitely cold-workable and printable – I can say that from experience. High silicon steel is problematic, 6.5\% is far more brittle than 3%.

Thank you for your comment. We have corrected this mistake.

Line 219:

The printability of Fe-Si alloys becomes problematic somewhere around 6.5wt\% silicon [73].

  • Line 618 “However, the Preisach or play model-based calculations seem the most promising iron loss models” – analyse the methods in the end of chapter 4, provide the reader with some kind of a comparison and then, based on that, make the conclusions and state which models are the best suited or promising for 3D printed materials. Then chapter 4.4.1 would make more sense – to show the reader how to best implement the identified method.

Thank you for your comment. We extended the introducing section of section with additional information and added some more insights to highlight the complexity of using Preisach-like models for such a simple use case. We highlight at the beginning of the section that this kind of methodology can consider the non-linearity of the bh-curve in the finite element level instead of the loss separation or Steinmetz methods. However, convergence can be a problem in complex use cases.

The modified text at the beginning of the section:

The Steinmetz equation or the loss separation-based formulas used at the post-processing stage of finite element simulations. Therefore, during the calculations, these methods can not accurately consider the non-linear \textcolor{red}{BH} characteristics in every calculated finite element. 
This section highlights how complex the Preisach model can be applied in FEM simulations to get the losses inside such a simple arrangement, i.e. inside a lamination made of material M250-35A.
The simple illustration shows that applying the Preisach model results in accurate loss calculation; however, if the anomalous losses term extends it, it results in a time-consuming algorithm. In this case, getting a convergent algorithm can be difficult for a more complex geometry. However, the calculated hysteresis loops at every finite element cell of the arrangement can be more exact,  which can benefit custom-designed electrical machines from SMLC materials.   Two kinds of Preisach-type models are applied in this section for a simple lamination to estimate the losses of the selected M250-35A grade steel and demonstrate the applicability of the Preisach models.

Round 3

Reviewer 2 Report

The authors have addressed the most pressing concerns.

-